# Causal Inference on Time Series using Restricted Structural Equation Models

**Jonas Peters**[*]
Seminar for Statistics
ETH Zürich, Switzerland
peters@math.ethz.ch

**Dominik Janzing**
MPI for Intelligent Systems
Tübingen, Germany
janzing@tuebingen.mpg.de

**Bernhard Schölkopf**
MPI for Intelligent Systems
Tübingen, Germany
bs@tuebingen.mpg.de

## Abstract

Causal inference uses observational data to infer the causal structure of the data generating system. We study a class of restricted Structural Equation Models for time series that we call Time Series Models with Independent Noise (TiMINo). These models require independent residual time series, whereas traditional methods like Granger causality exploit the variance of residuals. This work contains two main contributions: (1) *Theoretical:* By restricting the model class (e.g. to additive noise) we provide general identifiability results. They cover lagged and instantaneous effects that can be nonlinear and unfaithful, and non-instantaneous feedbacks between the time series. (2) *Practical:* If there are no feedback loops between time series, we propose an algorithm based on non-linear independence tests of time series. We show empirically that when the data are causally insufficient or the model is misspecified, the method avoids incorrect answers. We extend the theoretical and the algorithmic part to situations in which the time series have been measured with different time delays. TiMINo is applied to artificial and real data and code is provided.

## 1   Introduction

We first introduce the problem of causal inference on **iid data**, that is in the case with no time structure. Let therefore $X^i$, $i \in V$, be a set of random variables and let $\mathcal{G}$ be a directed acyclic graph (DAG) on $V$ describing the causal relationships between the variables. Given iid samples from $\mathbf{P}^{(X^i), i \in V}$, we aim at estimating the underlying causal structure of the variables $X^i, i \in V$. *Constraint- or independence-based methods* [e.g. Spirtes et al., 2000] assume that the joint distribution is Markov, and faithful with respect to $\mathcal{G}$. The PC algorithm, for example, exploits conditional independences for reconstructing the Markov equivalence class of $\mathcal{G}$ (some edges remain undirected). We say $\mathbf{P}^{(X^i), i \in V}$ satisfies a *Structural Equation Model* [Pearl, 2009] w.r.t. DAG $\mathcal{G}$ if for all $i \in V$ we can write $X^i = f_i(\mathbf{PA}^i, N^i)$, where $\mathbf{PA}^i$ are the parents of node $i$ in $\mathcal{G}$. Additionally, we require $(N^i)_{i \in V}$ to be jointly independent. By restricting the function class one can identify the bivariate case: Shimizu et al. [2006] show that if $\mathbf{P}^{(X,Y)}$ allows for $Y = a \cdot X + N_Y$ with $N_Y \perp\!\!\!\perp X$ then $\mathbf{P}^{(X,Y)}$ only allows for $X = b \cdot Y + N_X$ with $N_X \perp\!\!\!\perp Y$ if $(X, N_Y)$ are jointly Gaussian ( $\perp\!\!\!\perp$ stands for statistical independence). This idea has led to the extensions of nonlinear additive functions $f(x, n) = g(x) + n$ [Hoyer et al., 2009]. Peters et al. [2011b] show how identifiability for two variables generalizes to the multivariate case.

We now turn to the case of **time series data**. For each $i$ from a finite $V$, let therefore $\left(X_t^i\right)_{t \in \mathbb{N}}$ be a time series. $\mathbf{X}_t$ denotes the vector of time series values at time $t$. We call the infinite graph that contains each variable $X_t^i$ as a node the *full time graph*. The *summary time graph* contains all $\#V$

---

[*]Significant parts of this research was done, when Jonas Peters was at the MPI Tübingen.

components of the time series as vertices and an arrow between $X^i$ and $X^j$, $i \neq j$, if there is an arrow from $X^i_{t-k}$ to $X^j_t$ in the full time graph for some $k$. We are given a sample $(\mathbf{X}_1, \ldots, \mathbf{X}_T)$ of a multivariate time series and estimate the true summary time graph. I.i.d. methods are not directly applicable because a common history might introduce complicated dependencies between contemporaneous data $X_t$ and $Y_t$. Nevertheless several methods dealing with time series data are motivated by the iid setting (Section 2). Many of them encounter similar problems: when the model assumptions are violated (e.g. in the presence of a confounder) the methods draw false causal conclusions. Furthermore, they do not include nonlinear instantaneous effects. In this work, we extend the Structural Equation Model framework to time series data and call this approach *time series models with independent noise (TiMINo)*. These models include nonlinear and instantaneous effects. They assume $X_t$ to be a function of all direct causes and some noise variable, the collection of which is supposed to be jointly independent. This model formulation comes with substantial benefits: In Section 3 we prove that for TiMINo models the full causal structure can be recovered from the distribution. Section 4 introduces an algorithm (*TiMINo causality*) that recovers the model structure from a finite sample. It can be equipped with any algorithm for fitting time series. If the data do not satisfy the model assumptions, TiMINo causality remains mostly undecided instead of drawing wrong causal conclusions. Section 5 deals with time series that have been shifted by different (unknown) time delays. Experiments on simulated and real data sets are shown in Section 6.

## 2 Existing methods

**Granger causality** [Granger, 1969] (**G-causality** for the remainder of the article) is based on the following idea: $X^i$ does not Granger cause $X^j$ if including the past of $X^i$ does not help in predicting $X^j_t$ given the past of all all other time series $X^k$, $k \neq i$. In principle, "all other" means all other information in the world. In practice, one is limited to $X^k$, $k \in V$. The phrase "does not help" is translated into a significance test assuming a multivariate time series model. If the data follow the assumed model, e.g. the VAR model below, G-causality is sometimes interpreted as testing whether $X^i_{t-h}, h > 0$ is independent of $X^j_t$ given $X^k_{t-h}$, $k \in V \setminus \{i\}, h > 0$ [see Florens and Mouchart, 1982, Eichler, 2011, Chu and Glymour, 2008, Quinn et al., 2011, and ANLTSM below]. **Linear G-causality** considers a VAR model: $\mathbf{X}_t = \sum_{\tau=1}^p \mathbf{A}(\tau)\mathbf{X}_{t-\tau} + \mathbf{N}_t$, where $\mathbf{X}_t$ and $\mathbf{N}_t$ are vectors and $\mathbf{A}(\tau)$ are matrices. For checking whether $X^i$ G-causes $X^j$ one fits a full VAR model $M_{\text{full}}$ to $\mathbf{X}_t$ and a VAR model $M_{\text{restr}}$ to $\mathbf{X}_t$ that predicts $X^i_t$ without using $X^j$ (using the constraints $A_{\cdot i}(\tau) = 0$ for all $1 \leq \tau \leq p$). One tests whether the reduction of the residual sum of squares (RSS) of $X^i_t$ is significant by using the following test statistic: $T := \frac{(RSS_{\text{restr}} - RSS_{\text{full}})/(p_{\text{full}} - p_{\text{restr}})}{RSS_{\text{full}}/(N - p_{\text{full}})}$, where $p_{\text{full}}$ and $p_{\text{restr}}$ are the number of parameters in the respective models. For the significance test we use $T \sim F_{p_{\text{full}} - p_{\text{restr}}, N - p_{\text{full}}}$. G-causality has been extended to **nonlinear G-causality**, [e.g. Chen et al., 2004, Ancona et al., 2004]. In this paper we focus on an extension for the bivariate case proposed by Bell et al. [1996]. It is based on generalized additive models (gams) [Hastie and Tibshirani, 1990]: $X^i_t = \sum_{\tau=1}^p \sum_{j=1}^n f_{i,j,\tau}(X^j_{t-\tau}) + N^i_t$, where $\mathbf{N}_t$ is a $\#V$ dimensional noise vector. Bell et al. [1996] utilize the same $F$ statistic as above using estimated degrees of freedom.
Following Bell et al. [1996], Chu and Glymour [2008] introduce additive nonlinear time series models (**ANLTSM** for short) for performing relaxed conditional independence tests: If including one variable, e.g. $X^1_{t-1}$, into a model for $X^2_t$ that already includes $X^2_{t-2}, X^2_{t-1}$, and $X^1_{t-2}$ does not improve the predictability of $X^2_t$, then $X^1_{t-1}$ is said to be independent of $X^2_t$ given $X^2_{t-2}, X^2_{t-1}, X^1_{t-2}$ (if the maximal time lag is 2). Chu and Glymour [2008] propose a method based on constraint-based methods like FCI [Spirtes et al., 2000] in order to infer the causal structure exploiting those conditional independence statements. The instantaneous effects are assumed to be linear and the confounders linear and instantaneous.
**TS-LiNGAM** [Hyvärinen et al., 2008] is based on LiNGAM [Shimizu et al., 2006] from the iid setting. It allows for instantaneous effects and assumes all relationships to be linear.

These approaches encounter some methodological problems. *Instantaneous effects:* G-causality cannot deal with instantaneous effects. E.g., when $X_t$ is causing $Y_t$, including any of the two time series helps for predicting the other and G-causality infers $X \rightarrow Y$ and $Y \rightarrow X$. ANLTSM and TS-LiNGAM only allow for linear instantaneous effects. Theorem 1 shows that the summary time graph may still be identifiable when the instantaneous effects are linear and the variables are jointly Gaussian. TS-LiNGAM does not work in these situations. *Confounders:* G-causality might fail

when there is a confounder between $X_t$ and $Y_{t+1}$, say. The path between $X_t$ and $Y_{t+1}$ cannot be blocked by conditioning on any observed variables; G-causality infers $X \to Y$. We will see empirically that TiMINo remains undecided instead; Entner and Hoyer [2010] and Janzing et al. [2009] provide (partial) results for the iid setting. ANLTSM does not allow for nonlinear confounders or confounders with time structure and TS-LiNGAM may fail, too (Exp. 1). *Robustness:* Theorem 1 (ii) shows that performing general conditional independence tests suffices. The conditioning sets, however, are too large and the tests are performed under a simple model (e.g. VAR). If the model is misspecified, one may draw wrong conclusions without noticing (e.g. Exp. 3).

For TiMINo (defined below), Lemma 1 shows that after fitting and checking the model by using *un*conditional independence tests, the difficult conditional independences have been checked implicitly. A model check is not new [e.g. Hoyer et al., 2009, Entner and Hoyer, 2010] but is thus an effective tool. We can equip bivariate G-causality with a test for cross-correlations; this is not straight-forward for multivariate G-causality. Furthermore, using cross-correlation as an independence test does not always suffice (see Section 2).

## 3 Structural Equation models for time series: TiMINo

**Definition 1** *Consider a time series* $\mathbf{X}_t = (X_t^i)_{i \in V}$ *whose finite dimensional distributions are absolutely continuous w.r.t a product measure (e.g. there is a pdf or pmf). The time series satisfies a* TiMINo *if there is a* $p > 0$ *and* $\forall i \in V$ *there are sets* $\mathbf{PA}_0^i \subseteq X^{V \setminus \{i\}}, \mathbf{PA}_k^i \subseteq X^V$, *s.t.* $\forall t$

$$X_t^i = f_i\big((\mathbf{PA}_p^i)_{t-p}, \ldots, (\mathbf{PA}_1^i)_{t-1}, (\mathbf{PA}_0^i)_t, N_t^i\big), \tag{1}$$

*with* $N_t^i$ *jointly independent over* $i$ *and* $t$ *and for each* $i$, $N_t^i$ *are identically distributed in* $t$. *The corresponding full time graph is obtained by drawing arrows from any node that appears in the right-hand side of* (1) *to* $X_t^i$. *We require the full time graph to be acyclic. Section 6 shows examples.*

Theorem 1 (i) assumes that (1) follows an identifiable functional model class (IFMOC). This means that (I) *causal minimality* holds, a weak form of faithfulness that assumes a statistical dependence between cause and effect given all other parents [Spirtes et al., 2000]. And (II), all $f_i$ come from a function class that is small enough to make the bivariate case identifiable. Peters et al. [2011b] give a precise definition. Important examples include nonlinear functions with additive Gaussian noise and linear functions with additive non-Gaussian noise. Due to space constraints, proofs are provided in the appendix. In the one-dimensional linear case model (1) is time-reversible if and only if the noise is normally distributed [Peters et al., 2009].

**Theorem 1** *Suppose that* $\mathbf{X}_t$ *can be represented as a TiMINo* (1) *with* $\mathbf{PA}(X_t^i) = \bigcup_{k=0}^p (\mathbf{PA}_k^i)_{t-k}$ *being the direct causes of* $X_t^i$ *and that one of the following holds:*

   *(i) Equations* (1) *come from an IFMOC (e.g. nonlinear functions* $f_i$ *with additive Gaussian noise* $N_t^i$ *or linear functions* $f_i$ *with additive non-Gaussian noise* $N_t^i$*). The summary time graph can contain cycles.*

   *(ii) Each component exhibits a time structure (*$\mathbf{PA}(X_t^i)$ *contains at least one* $X_{t-k}^i$*), the joint distribution is faithful w.r.t. the full time graph, and the summary time graph is acyclic.*

*Then the full time graph can be recovered from the joint distribution of* $\mathbf{X}_t$. *In particular, the true causal summary time graph is identifiable. (Neither of the conditions (i) and (ii) implies the other.)*

Many function classes satisfy (i) [Peters et al., 2013]. To estimate $f_i$ from data ($\mathbf{E}[X_t^i | \mathbf{X}_{t-p}, \ldots, \mathbf{X}_{t-1}]$ for additive noise) we require stationarity and/or $\alpha$ mixing, or geometric ergodicity [e.g. Chu and Glymour, 2008]. Condition (ii) shows how time structure simplifies the causal inference problem. For iid data the true graph is not identifiable in the linear Gaussian case; with time structure it is. We believe that condition (ii) is more difficult to verify in practice; faithfulness is not required for (i). In (ii), the acyclicity prevents the full time graph from being fully connected up to order $p$.

## 4 A practical method: TiMINo causality

The algorithm for TiMINo causality is based on the theoretical finding in Theorem 1. It takes the time series data as input and outputs either a DAG that estimates the summary time graph or remains undecided. It tries to fit a TiMINo model to the data and outputs the corresponding graph. If

no model with independent residuals is found, it outputs "I do not know". This becomes intractable for a time series with many components; for time series without feedback loops, we adapt a method for additive noise models without time structure suggested by Mooij et al. [2009] that avoids enumerating all DAGs. Algorithm 1 shows the modified version. As reported by Mooij et al. [2009], the time complexity is $\mathcal{O}(d^2 \cdot f(n, d) \cdot t(n, d))$, where $d$ is the number of time series, $n$ the sample size and $f(n, d)$ and $t(n, d)$ the complexity of the user-specific regression method and independence test, respectively. Peters et al. [2013] discuss the algorithm's correctness. We present our choices but do not claim their optimality, any other fitting method and independence test can be used, too.

---

**Algorithm 1** TiMINo causality

---

1: **Input:** Samples from a $d$-dimensional time series of length $T$: $(\mathbf{X}_1, \ldots, \mathbf{X}_T)$, maximal order $p$
2: $S := (1, \ldots, d)$
3: **repeat**
4:     **for** $k$ in S **do**
5:         Fit TiMINo for $X_t^k$ using $X_{t-p}^k, \ldots, X_{t-1}^k, X_{t-p}^i, \ldots, X_{t-1}^i, X_t^i$ for $i \in S \setminus \{k\}$
6:         Test if residuals are indep. of $X^i, i \in S$.
7:     **end for**
8:     Choose $k^*$ to be the $k$ with the weakest dependence. (If there is no $k$ with independence, break and output: "I do not know - bad model fit").
9:     $S := S \setminus \{k^*\};$   $\mathrm{pa}(k^*) := S$
10: **until** length($S$)$= 1$
11: For all $k$ remove all parents that are not required to obtain independent residuals.
12: **Output:** $(\mathrm{pa}(1), \ldots, \mathrm{pa}(\mathrm{d}))$

---

Depending on the assumed model class, TiMINo causality has to be provided with a **fitting method**. Here, we chose the R functions `ar` for VAR fitting ($f_i(p_1, \ldots, p_r, n) = a_{i,1} \cdot p_1 + \ldots + a_{i,r} \cdot p_r + n$), `gam` for generalized additive models ($f_i(p_1, \ldots, p_r, n) = f_{i,1}(p_1) + \ldots + f_{i,r}(p_r) + n$) [e.g. Bell et al., 1996] and `gptk` for GP regression ($f_i(p_1, \ldots, p_r, n) = f_i(p_1, \ldots, p_r) + n$). We call the methods TiMINo-linear, TiMINo-gam and TiMINo-GP, respectively. For the first two AIC determines the order of the process. All fitting methods are used in a "standard way". For `gam` we used the built-in nonparametric smoothing splines. For the GP we used zero mean, squared exponential covariance function and Gaussian Likelihood. The hyper-parameters are automatically chosen by marginal likelihood optimization. Code is available online.

To **test for independence** between a residual time series $N_t^k$ and another time series $X_t^i, i \in S$, we shift the latter time series up to the maximal order $\pm p$ (but at least up to $\pm 4$); for each of those combinations we perform HSIC [Gretton et al., 2008], an independence test for iid data. One could also use a test based on cross-correlation that can be derived from Thm 11.2.3. in [Brockwell and Davis, 1991]. This is related to what is done in transfer function modeling [e.g. §13.1 in Brockwell and Davis, 1991], which is restricted to two time series and linear functions. As opposed to the iid setting, testing for cross-correlation is often enough in order to reject a wrong model. Only Experiments 1 and 5 describe situations, in which cross-correlations fail. To reduce the running time one can use cross-correlation to determine the graph structure and use HSIC as a final model check. For HSIC we used a Gaussian kernel; as in [Gretton et al., 2008], the bandwidth is chosen such that the median distance of the input data leads to an exponent of one. Testing for non-vanishing autocorrelations in the residuals is not included yet.

If the model assumptions only hold in some parts of the summary time graph, we can still try to **discover parts of the causal structure**. Our code package contains this option. We obtained positive results on simulated data but there is no corresponding identifiability statement.

Our method has some **potential weaknesses**. It can happen that one is able to fit a model only in the wrong direction. This, however, requires an "unnatural" fine tuning of the functions [Janzing and Steudel, 2010] and is relevant only when there are time series without time structure or the data are non-faithful (see Theorem 1). The null hypothesis of the independence test represents independence, although the scientific discovery of a causal relationship should rather be the alternative hypothesis. This fact may lead to wrong causal conclusions (instead of "I do not know") on small data sets. The effect is strengthened by the Bonferroni correction of the HSIC based independence test; one may require modifications for a high number of time series components. For large sample sizes, even

smallest differences between the true data generating process and the model may lead to rejected independence tests [discussed by Peters et al., 2011a].

# 5 TiMINo for Shifted Time Series

In some applications, we observe the components of the time series with varying time delay. Instead of $X_t^i$ we are then working with $\tilde{X}_t^i = X_{t-\ell}^i$, with $0 \leq \ell \leq k$. E.g., in functional magnetic resonance imaging brain activity is measured through an increased blood flow in the corresponding area. It has been reported that these data often suffer from different time delays [e.g. Buxton et al., 1998, Smith et al., 2011]. Given the (shifted) measurements $\tilde{X}_t^i$, we therefore have to cope with causal relationships that go backward in time. This is only resolved when going back to the unobserved true data $X_t^i$. Measures like Granger causality will fail in these situations. This does not necessarily have to be the case, however. The structure still remains identifiable even if we observe $\tilde{X}_t^i$ instead of $\tilde{X}_t^i$ (the following theorem generalizes the second part of Theorem 1 and is proved accordingly)[1]:

**Theorem 2** *Assume condition (ii) from Theorem 1 with $\tilde{X}_t^i = X_{t-\ell}^i$, where $0 \leq \ell \leq k$ are unknown time delays. Then, the full time graph of $\tilde{\mathbf{X}}_t$ is identifiable from the joint distribution of $\tilde{\mathbf{X}}_t$. In particular, the summary time graphs of $\tilde{\mathbf{X}}_t$ and $\mathbf{X}_t$ are identical and therefore identifiable.*

As opposed to Theorem 1 we cannot identify the full time graph of $\mathbf{X}_t$. It may not be possible, for example, to distinguish between a lag two effect from $X^1$ to $X^2$ and a corresponding lag one effect with a shifted time series $X^2$. The method for recovering the network structure stays almost the same as the one for non-shifted time series. only line 5 of Algorithm 1 has to be updated: we additionally include $X_{t+\ell}^i$ for $0 \leq \ell \leq k$ for all $i \in S \setminus \{k\}$. While TiMINo exploits an asymmetry between cause and effect emerging from restricted structural equations, G-causality exploits the asymmetry of time. The latter asymmetry is broken when considering shifted time series.

# 6 Experiments

## 6.1 Artificial Data

We always included instantaneous effects, fitted models up to order $p = 2$ or $p = 6$ and set $\alpha = 0.05$.

**Experiment 1: Confounder with time lag.** We simulate 100 data sets (length 1000) from $Z_t = a \cdot Z_{t-1} + N_{Z,t}$, $X_t = 0.6 \cdot X_{t-1} + 0.5 \cdot Z_{t-1} + N_{X,t}$, $Y_t = 0.6 \cdot Y_{t-1} + 0.5 \cdot Z_{t-2} + N_{Y,t}$, with $a$ between 0 and 0.95 and $N_{\cdot,t} \sim 0.4 \cdot \mathcal{N}(0,1)^3$. Here, $Z$ is a hidden common cause for $X$ and $Y$. For all $a$, $X_t$ contains information about $Z_{t-1}$ and $Y_{t+1}$ (see Figure 1); G-causality and TS-LiNGAM wrongly infer $X \to Y$. For large $a$, $Y_t$ contains additional information about $X_{t+1}$, which leads to the wrong arrow $Y \to X$. TiMINo causality does not decide for any $a$. The nonlinear methods perform very similar (not shown). For $a = 0$, a cross-correlation test is not enough to reject $X \to Y$. Further, all methods fail for $a = 0$ and Gaussian noise. (Similar results for non-linear confounder.)

**Experiment 2: Linear, Gaussian with instantaneous effects.** We sample 100 data sets (length 2000) from $X_t = A_1 \cdot X_{t-1} + N_{X,t}$, $W_t = A_2 \cdot W_{t-1} + A_3 \cdot X_t + N_{W,t}$, $Y_t = A_4 \cdot Y_{t-1} + A_5 \cdot W_{t-1} + N_{Y,t}$, $Z_t = A_6 \cdot Z_{t-1} + A_7 \cdot W_t + A_8 \cdot Y_{t-1} + N_{Z,t}$ and $N_{\cdot,t} \sim 0.4 \cdot \mathcal{N}(0,1)$ and $A_i$ iid from $\mathcal{U}([-0.8, -0.2] \cup [0.2, 0.8])$. We regard the graph containing $X \to W \to Y \to Z$ and $W \to Z$ as correct. TS-LiNGAM and G-causality are not able to recover the true structure (see Table 1). We obtain similar results for non-linear instantaneous interactions.

**Experiment 3: Nonlinear, non-Gaussian without instantaneous effects.** We simulate 100 data sets (length 500) from $X_t = 0.8 X_{t-1} + 0.3 N_{X,t}$, $Y_t = 0.4 Y_{t-1} + (X_{t-1} - 1)^2 + 0.3 N_{Y,t}$, $Z_t = 0.4 Z_{t-1} + 0.5 \cos(Y_{t-1}) + \sin(Y_{t-1}) + 0.3 N_{Z,t}$, with $N_{\cdot,t} \sim \mathcal{U}([-0.5, 0.5])$ (similar results for other noise distributions, e.g. exponential). Thus, $X \to Y \to Z$ is the ground truth. Nonlinear G-causality fails since the implementation is only pairwise and it thus always infers an effect from $X$ to $Z$. Linear G-causality cannot remove the nonlinear effect from $X_{t-2}$ to $Z_t$ by using $Y_{t-1}$. Also TiMINo-linear assumes a wrong model but does not make any decision. TiMINo-gam and TiMINo-GP work well on this data set (Table 2). This specific choice of parameters show that a significant

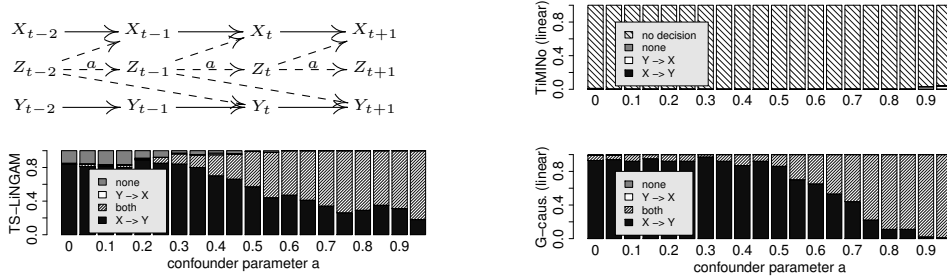

Figure 1: Exp.1: Part of the causal full time graph with hidden common cause $Z$ (top left). TiMINo causality does not decide (top right), whereas G-causality and TS-LiNGAM wrongly infer causal connections between $X$ and $Y$ (bottom).

| DAG | G-causal. linear | TiMINo linear | TS-LiNGAM |
|---|---|---|---|
| correct | 13% | 83% | 19% |
| wrong | 87% | 7% | 81% |
| no dec. | 0% | 10% | 0% |

Table 1: Exp.2: Gaussian data and linear instantaneous effects: only TiMINo mostly discovers the correct DAG.

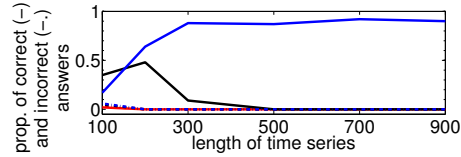

Figure 2: Exp.4: TiMINo-GP (blue) works reliably for long time series. TiMINo-linear (red) and TiMINo-gam (black) mostly remain undecided.

difference in performance is possible. For other parameters (e.g. less impact of the nonlinearity), G-causality and TS-LiNGAM still assume a wrong model but make fewer mistakes.

Table 2: Exp.3: Since the data are nonlinear, linear G-causality and TS-LiNGAM give wrong answers, TiMINo-lin does not decide. Nonlinear G-causality fails because it analyzes the causal structure between pairs of time series.

| DAG | Granger$_{lin}$ | Granger$_{nonlin}$ | TiMINo$_{lin}$ | TiMINo$_{gam}$ | TiMINo$_{GP}$ | TS-LiNGAM |
|---|---|---|---|---|---|---|
| correct | 69% | 0% | 0% | 95% | 94% | 12% |
| wrong | 31% | 100% | 0% | 1% | 1% | 88% |
| no dec. | 0% | 0% | 100% | 4% | 5% | 0% |

**Experiment 4: Non-additive interaction.** We simulate 100 data sets with different lengths from $X_t = 0.2 \cdot X_{t-1} + 0.9 N_{X,t}, Y_t = -0.5 + \exp(-(X_{t-1} + X_{t-2})^2) + 0.1 N_{Y,t}$, with $N_{\cdot,t} \sim \mathcal{N}(0,1)$. Figure 2 shows that TiMINo-linear and TiMINo-gam remain mainly undecided, whereas TiMINo-GP performs well. For small sample sizes, one observes two effects: GP regression does not obtain accurate estimates for the residuals, these estimates are not independent and thus TiMINo-GP remains more often undecided. Also, TiMINo-gam makes more correct answers than one would expect due to more type II errors. Linear G-causality and TS-LiNGAM give more than $90\%$ incorrect answers, but non-linear G-causality is most often correct (not shown). Bad model assumptions do not *always* lead to incorrect causal conclusions.

**Experiment 5: Non-linear Dependence of Residuals.** In Experiment 1, TiMINo equipped with a cross-correlation inferred a causal edge, although there were none. The opposite is also possible: $X_t = -0.5 \cdot X_{t-1} + N_{X,t}, Y_t = -0.5 \cdot Y_{t-1} + X_{t-1}^2 + N_{Y,t}$ and $N_{\cdot,t} \sim 0.4 \cdot \mathcal{N}(0,1)$ (length 1000). TiMINo-gam with cross-correlation infers no causal link between $X$ and $Y$, whereas TiMINo-gam with HSIC correctly identifies $X \to Y$.

**Experiment 6: Shifted Time Series.** We simulate 100 random DAGs with $\#V = 3$ nodes by choosing a random ordering of the nodes and including edges with a probability of $0.6$. The structural equations are additive (gam). Each component is of the form $f(x) = a \cdot \max(x, -0.1) + b \cdot \mathrm{sign}(x)\sqrt{|x|}$, with $a, b$ iid from $\mathcal{U}([-0.5, -0.2] \cup [0.2, 0.5])$. We sample time series (length 1000) from Gaussian noise and observe the sink node time series with a time delay of three. This makes all

traditional methods inapplicable. The performance of linear G-causality, for example, drops from an average Structural Hamming Distance (SHD) of $0.38$ without time delay to $1.73$ with time delay. TiMINo-gam causality recognizes the wrong model assumption. The SHD increases from $0.13$ (17 undecided cases) to $0.71$ (79 undecided cases). Adjusting for a time delay (Section 5) yields an SHD of $0.70$ but many more decisions (18 undecided cases). Although it is possible to adjust for time delays, the procedure enlarges the model space, which makes rejecting all wrong models more difficult. Already $\#V = 5$ leads to worse average SHD: G-causality: $4.5$, TiMINo-gam: $1.5$ (92 undecided cases) and TiMINo-gam with time delay adjustment: $2.4$ (38 undecided cases).

## 6.2   Real Data

We fitted up to order $6$ and included instantaneous effects. For TiMINo, "correct" means that TiMINo-gam is correct and TiMINo-linear is correct or undecided. TiMINo-GP always remains undecided because there are too few data points to fit such a general model. Again, $\alpha$ is set to $0.05$.

**Experiment 7: Gas Furnace.** [Box et al., 2008, length 296], $X_t$ describes the input gas rate and $Y_t$ the output $CO_2$. We regard $X \rightarrow Y$ as being true. TS-LiNGAM, G-causality, TiMINo-lin and TiMINo-gam correctly infer $X \rightarrow Y$. Disregarding time information leads to a wrong causal conclusion: The method described by Hoyer et al. [2009] leads to a $p$-value of $4.8\%$ in the correct and $9.1\%$ in the false direction.

**Experiment 8: Old Faithful.** [Azzalini and Bowman, 1990, length 194] $X_t$ contains the duration of an eruption and $Y_t$ the time interval to the next eruption of the Old Faithful geyser. We regard $X \rightarrow Y$ as the ground truth. Although the time intervals $[t, t+1]$ do not have the same length for all $t$, we model the data as two time series. TS-LiNGAM and TiMINo give correct answers, whereas linear G-causality infers $X \rightarrow Y$, and nonlinear G-causality infers $Y \rightarrow X$.

**Experiment 9: Abalone (no time structure).** The abalone data set [Asuncion and Newman, 2007] contains (among others that lead to similar results) age $X_t$ and diameter $Y_t$ of a certain shell fish. If we model 1000 randomly chosen samples as time series, G-causality (both linear and nonlinear) infers no causal relation as expected. TS-LiNGAM wrongly infers $Y \rightarrow X$, which is probably due to the nonlinear relationship. TiMINo gives the correct result.

**Experiment 10: Diary (confounder).** We consider 10 years of weekly prices for butter $X_t$ and cheddar cheese $Y_t$ (length 522, http://future.aae.wisc.edu/tab/prices.html) We expect their strong correlation to be due to the (hidden) milk price $M_t$: $X \leftarrow M \rightarrow Y$. TiMINo does not decide, whereas TS-LiNGAM and G-causality wrongly infer $X \rightarrow Y$. This may be due to different time lags of the confounder (cheese has longer storing and maturing times than butter).

**Experiment 11: Temperature in House.** We placed temperature sensors in six rooms (1 - Shed, 2 - Outside, 3 - Kitchen Boiler, 4 - Living Room, 5 - WC, 6 - Bathroom) of a house in the black forest and recorded the temperature on an hourly basis (length 7265). This house is not inhabited for most of the year, and lacking central heating; the few electric radiators start if the temperatur drops close to freezing. TiMINo does not decide since the model leads to dependent residuals. Although we do not provide any theory for the following steps, we analyze the model leading to the "least dependent" residuals by setting the test level $\alpha$ to zero. TiMINo causality then outputs a causal ordering of the variables. We applied TiMINo-lin and TiMINo-gam to the data sets using lags up to twelve (half a day) and report the proportion in which node $i$ precedes node $j$ (see matrix). This procedure reveals a sensible causal structure (we - arbitrarily- refer to entries larger than $0.5$ as causation). 2 (outside) causes all other readings, and none of the other temperatures causes 2. 1 (shed) causes all other readings except for 2. This is wrong, but not surprising since the shed's temperature is rather close to the outside temperature. 4 (living room) does not cause any other reading, but every other reading does cause it (the living room is

$$\begin{pmatrix} 0 & 0.25 & 0.83 & 1 & 1 & 1 \\ 0.75 & 0 & 0.83 & 1 & 1 & 1 \\ 0.17 & 0.17 & 0 & 0.75 & 0.33 & 0.33 \\ 0 & 0 & 0.25 & 0 & 0 & 0 \\ 0 & 0 & 0.67 & 1 & 0 & 0 \\ 0 & 0 & 0.67 & 1 & 1 & 0 \end{pmatrix}$$

the only room without any heating). The links $5 \rightarrow 3$ and $6 \rightarrow 3$ appear spurious, and come with numbers close to 0.5. These might be erroneous, however, they might also be due to the fact that sensor 3 is sitting on top of the kitchen boiler, which acts as a heat reservoir that delays temperature changes. The link $6 \rightarrow 5$ comes with a large number, but it is plausible since unlike the WC, the

bathroom has a window and is thus affected directly by outside temperature, causing fast regulation of its radiator, which is placed on a thin wooden wall facing the WC.

The phase slope index [Nolte et al., 2008] performed well in Exp. 7, in all other experiments it either gave wrong results or did not decide. Due to space constraints we omit details about this method. We did not find any code for ANLTSM.

## 7    Conclusions and Future Work

This paper shows how causal inference on time series benefits from the framework of Structural Equation Models. The identifiability statement is more general than existing results. The algorithm is based on unconditional independence tests and is applicable to multivariate, linear, nonlinear and instantaneous interactions. It contains the option of remaining undecided. While methods like Granger causality are built on the asymmetry of time direction, TiMINo additionally takes into account identifiability emerging from restricted structural equation models. This leads to a straightforward way of dealing with (unknown) time delays in the different time series. Although an extensive evaluation on real data sets is still required, we believe that our results emphasize the potential use of causal inference methods. They may provide guidance for future interventional experiments.

As future work one may use heteroscedastic models [Chen et al., 2012] and systematically preprocess the data (removing trends, periodicities, etc.). This may reduce the number of cases where TiMINo causality is undecided. TiMINo causality evaluates a model fit by checking independence of the residuals. As suggested in Mooij et al. [2009], Yamada and Sugiyama [2010], one may make the independence of the residuals as a criterion for the fitting process or at least for order selection.

## 8    Appendix

**Lemma 1 (Markov Condition for TiMINo)** *If $\mathbf{X}_t = (X_t^i)_{i \in V}$ satisfy a TiMINo model, each variable $X_t^i$ is conditionally independent of each of its non-descendants given its parents.*

**Proof .** With $\mathcal{S} := \mathbf{PA}(X_t^i) = \bigcup_{k=0}^{p}(\mathbf{PA}_k^i)_{t-k}$ and Eq. (1) we get $X_t^i|_{\mathcal{S}=s} = f_i(s, N_t^i)$ for an $s$ with $p(s) > 0$. Any non-descendant of $X_t^i$ is a function of all noise variables from its ancestors and is thus independent of $X_t^i$ given $\mathcal{S} = s$. This is the only time we assume $t \in \mathbb{N}$ in this paper.     $\square$

**Proof of Theorem 1** Suppose that $\mathbf{X}_t$ allows for two TiMINo representations that lead to different full time graphs $\mathcal{G}$ and $\mathcal{G}'$. (i) Assume that $\mathcal{G}$ and $\mathcal{G}'$ do not differ in the instantaneous effects: $\mathbf{PA}_0^i(\text{in } \mathcal{G}) = \mathbf{PA}_0^i(\text{in } \mathcal{G}') \ \forall i$. Wlog, there is some $k > 0$ and an edge $X_{t-k}^1 \to X_t^2$, say, that is in $\mathcal{G}$ but not in $\mathcal{G}'$. From $\mathcal{G}'$ and Lemma 1 we have that $X_{t-k}^1 \perp\!\!\!\perp X_t^2 \,|\, \mathcal{S}$, where $\mathcal{S} = (\{X_{t-l}^i, 1 \le l \le p, i \in V\} \cup \mathbf{ND}_t) \setminus \{X_{t-k}^1, X_t^2\}$, and $\mathbf{ND}_t$ are all $X_t^i$ that are non-descendants (wrt instantaneous effects) of $X_t^2$. Applied to $\mathcal{G}$, causal minimality leads to a contradiction: $X_{t-k}^1 \not\perp\!\!\!\perp X_t^2 \,|\, \mathcal{S}$. Now, let $\mathcal{G}$ and $\mathcal{G}'$ differ in the instantaneous effects and choose $\mathcal{S} = \{X_{t-l}^i, 1 \le l \le p, i \in V\}$. For each $s$ and $i$ we have: $X_t^i|_{\mathcal{S}=s} = f_i(s, (\tilde{\mathbf{PA}}_0^i)_t)$, where $\tilde{\mathbf{PA}}_0^i$ are all instantaneous parents of $X_t^i$ conditioned on $\mathcal{S} = s$. All $X_t^i|_{\mathcal{S}=s}$ with the instantaneous effects describe two different structures of an IFMOC. This contradicts the identifiability results by Peters et al. [2011b]. (ii) Because of Lemma 1 and faithfulness $\mathcal{G}$ and $\mathcal{G}'$ only differ in the instantaneous effects. But each instantaneous arrow $X_t^i \to X_t^j$ forms a $v$-structure together with $X_{t-k}^j \to X_t^j$; $X_{t-k}^j$ cannot be connected with $X_t^i$ since this introduces a cycle in the summary time graph.     $\square$

**Proof of Theorem 2** Two full time graphs $\mathcal{G}$ and $\mathcal{G}'$ for $\tilde{\mathbf{X}}_t$ can differ only in the directions of edges between time series. Assume $X_t^i \to X_{t+k}^j$ in $\mathcal{G}$ and $X_t^i \leftarrow X_{t+k}^j$ in $\mathcal{G}'$. Choose the largest $k$ possible. Then there is a $v$-structure $X_{t-\ell}^i \to X_t^i \leftarrow X_{t+k}^j$ for some $\ell$. A connection between $X_{t-\ell}^i$ and $X_{t+k}^j$ would lead to a pair as above with a larger $k$.     $\square$

## Footnotes

[1]We believe that a corresponding statement for condition (i) holds, too.

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
