[Reviews · NeurIPS 2013]

Submitted by Assigned_Reviewer_1

I read the author feedbacks:

- Theorem 1 (i) (our main result) does not require acyclicity of the summary time graph. Note that acyclicity is only one of two sufficient conditions.

I found that I did not reflect enough that the acyclic assumption is one of two sufficient conditions. Thanks. I changed the score up.

I still don't understand why they tested their method only on data which are generated under the acyclic assumption on the summary time graph. That seems to be more limited cases.






Summary of the paper:
This paper considers a class of structural equation models for times series data.
The models allow nonlinear instantaneous effects and lagged effects. On the other hand, Granger-causality based methods do not allow instantaneous effects and a linear non-Gaussian method TS-LiNGAM (Hyvarinen et al., ICML2008, JMLR2010) assumes linear effects.

Then they gave Theorem 1, which shows the identifiability conditions of the model. The conditions are i) the functions that causally relates variables come from a class of functions called IFMOC considered in Peters et al. [2011b], ii) the faithfulness and iii) the summary time graph is acyclic (They define the summary time graph is a graph that contains all the nodes of the time series and an arrow X^i (X at time point i) and X^j, i ¥neq j, if there is an arrow from X^i_{t-k} to X_t^i in the full time graph for SOME k in lines 53-55). The theorem 1 is an extension of Peters et al. [2011b] to a time series case and a partial extension of Hyvarinen et al. (ICML2008, JMLR2010) to nonlinear cases. Hyvarinen et al. proves the identifiability of their model assuming linearity and non-Gaussian residuals (disturbances or external influences), in which the summary graph needs not to be acyclic.

They further presented an estimation algorithm (Algorithm 1), which is a natural extension of Mooij et al. [2009] to time series cases. They use independence test to detect possible violations of model assumptions. Their algorithm can be applied (with small modifications) to shifted time series cases, which are often observed in fMRI data.

They conducted artificial data experiments to test their algorithms under various settings including latent confounders with time lag but still assumed that the summary graph is acyclic. They compared their results with a Granger-causality-based linear method and a linear non-Gaussian method TS-LiNGAM (Hyvarinen et al., ICML2008). Their methods are better in some nonlinear cases.

Finally, they tested some real datasets. Their methods worked better in some cases and were comparable to linear non-Gaussian TS-LiNGAM in some cases.

Pros.:
- It is interesting to extend TS-LiNGAM (Hyvarinen et al., ICML2008; JMLR2010) to nonlinear cases.
- It is practically important to consider shifted time series cases.

Cons.:
- The assumption in Theorem 1 that the summary graph is acyclic seems too restrictive unless I miss something important. It means that it does not happen that X^1_t-1 causes X^2_t and X^2_t-1 causes X^1_t, doesn’t it??
They claim that their methods is a kind of safe methods since it outputs `I don’t know’ if independence between exploratory variables and residuals is statistically rejected, which implies some violations of the model assumptions. The idea is not very new. It can be found in many papers, e.g., Entner and Hoyer (AMBN2010), Tashiro et al. (ICANN2012), Shimizu and Kano (2008, JSPI) and Hoyer et al. (NIPS2009).

Suggestions:
Would it be possible to prove Theorem 1 without assuming that the summary graph is acyclic?

- They claim that their method is a kind of safe methods since it outputs `I don’t know’ if independence between exploratory variables and residuals is rejected. The idea is not very new. For example, they should at least prove that explanatory variables and residuals are not independent if latent confounders exist to make their contribution clearer. Entner and Hoyer (AMBN2010) and Tashiro et al. (ICANN2012) proved that in linear non-Gaussian cases.

Comments

An extension of TS-LiNGAM to cases with latent confounders with time lag might be
Kawahara, S. Shimizu, and T. Washio. Analyzing relationships among ARMA processes based on non-Gaussianity of external influences.Neurocomputing, 74(12-13): 2212-2221, 2011.
It considers ARMA instead of AR model to model temporal dependency. Another extension of TS-LiNGAM to cases with instantaneous confounders would be
Gao and H. Yang. Identifying structural VAR model with latent variables using overcomplete ICA. Far East Journal of Theoretical Statistics, 40(1): 31-44, 2012.
In the last year NIPS, TS-LiNGAM was extended to cases with variance-dependent external influences.
Z. Chen, K. Zhang, and L. Chan. Causal discovery with scale-mixture model for spatiotemporal variance dependencies. In Advances in Neural Information Processing Systems 25 (NIPS2012), pp. xx-xx, 2012.
Taking these developments in linear non-Gaussian models into account, the proposed nonlinear model assuming the sample time graph is acyclic might be a small contribution, although extending to nonlinear models certainly is an interesting topic.

In Experiment 11, they say that “TiMINo does not decide since the model leads to dependent residuals. We nevertheless analyzed ...” I didn’t follow why they decided to continue to analyze the data even if the residuals were found to be dependent.

- They say that “we now refer to entries larger than 0.5 as causation”. I didn’t follow why they choose the threshold 0.5.
Summary: It is interesting and important to extend a linear non-Gaussian time series causal model TS-LiNGAM (Hyvarinen et al., ICML2008; JMLR2010) to nonlinear cases.
However, Theorem~1, which would be the main contribution of this paper makes a too restrictive assumption that the summary time graph is acyclic, and Theorem~1 seems not very useful. This seems to make the other parts less attractive.

Submitted by Assigned_Reviewer_2

The authors discuss techniques to infer causal influence between time
series. Causal inference for time-series methods is an increasingly relevant
topic in many domains. The presentation is clean and they include a
number of examples (simulations and real data). This works appears as
a natural progression of some earlier causal inference methods for
i.i.d. variables.

The only major changes I would suggest are to compare this work with
Eichler’s work theoretically and experimentally and to compare this
work with Chu and Glymour’s experimentally. These points are
elaborated below.

The setting the authors consider involves time-series without
feedback. For many applications this could be a significant
restriction. There should be some discussion about this issue. It is
noteworthy that the algorithm (adapted from Mooij) does not require
knowing the ordering. However, there is an underlying assumption
though from Theorem 1 (ii) that the joint distribution is faithful
w.r.t. the full time graph. This might limit the range of nonlinear
relationships that the algorithm could identify.

In contrast, Eichler’s work (Granger causality method) can handle
feedback loops, though in handling instantaneous relationships it
consequently does not choose a direction. Like this work, Eichler’s
work uses independence tests. Those might perform better than the
linear and non-linear Granger causality methods currently used in the
experimental section.
More should be done to experimentally contrast the authors’ work with
that proposed by Chu and Glymour(“ANLTSM”). It is unclear what
differences are in run-time or performance between these algorithms.

In comparing this work with ANLTSM, the authors should include
examples with nonlinear instantaneous effects or nonlinear
confounders. In the paper, ANLTSM is mentioned as being more
restrictive than the authors’ work, as the former requires confounders
and instantaneous influences to be linear. However, none of the
simulations have nonlinear instantaneous effects or confounders.

It might be worthwhile to know of Yamada and
Sugiyama’s “Dependence Minimizing Regression with Model Selection for
Non-Linear Causal Inference under Non-Gaussian Noise” as a possible
alternative to the HSIC independence tests.

Some other missing references are "Estimating the Directed Information to Infer Causal Relationships in Ensemble Neural Spike Train Recordings" and "Equivalence between minimal generative model graphs and directed information graphs".
Summary: Authors tackle an important problem and they do a nice job presenting their results. The paper can be improved by including comparison with Eichler’s work theoretically and experimentally and by comparison with Chu and Glymour’s experimentally.

Submitted by Assigned_Reviewer_4

This paper introduces a model and procedure for learning instantaneous and lagged causal relationships among variables in a time series when each causal relationship is either identifiable in the sense of the additive noise model (Hoyer et al. 2009) or exhibits a time structure. The learning procedure finds a causal order by iteratively fitting VAR or GAM models where each variable is a function of all other variables and making the variable with the least dependence the lowest variable in the order. Excess parents are then pruned to produce the summary causal graph (where x->y indicates either an instantaneous or lagged cause up to the order of the VAR or GAM model that is fit). Experiments show that the method outperforms competing methods and returns no results in cases where the model can be identified (rather than wrong results).

This paper is an interesting extension to the additive noise model approach to learning causal relationships and the results look promising. However, I think the presentation/organization is at times unclear and leaves out important details.

I think the notation in (1) is unnecessarily confusing (including a subscript, superscript, and additional outer parentheses subscript for each parents set). I wasn't certain about what it actually meant until after reading on. I still don't see why the two separate subscripts are necessary to coherently define the model.

Rather than introducing the term IFMOC, which requires introducing a weak form of faithfulness (which many in the NIPS community may not be familiar with anyway) as well as referencing identifiability conditions discussed in another paper, and then mentioning it in theorem 1, I think it would be more clear and certainly more rigorous to just provide the explicit conditions for identifiability in theorem 1.

For algorithm 1, it is never discussed how extra parents are pruned from the graph (line 12). Is this done in the same way as Mooij et al. 2009? Does the time structure introduce any problems (other than if there are feedback loops)?

Also, is there a proof that algorithm 1 is correct? A proof or at least further discussion (even just specifying why a proof from Mooij et al. 2009 should go through without issues in this case) would be helpful.
Summary: The authors present an interesting approach to learning instantaneous and lagged causal relationships from time series data which promising empirical results, but the presentation and organization of the paper could be improved and some details for understanding the procedure are omitted.

Submitted by Assigned_Reviewer_5

The paper present two sets of assumptions under which a dynamic NPSEM may be recovered from multivariate time-series data.

The authors relate their approach to existing approaches based on Granger-causality.

There are two sets of assumptions provided. The method (and proof) based on the first set of assumptions seems to be building on the method of Peters et al., designed for iid data. The second set of assumptions essentially uses the fact that the time summary graph is assumed to be acyclic together with the fact that it is supposed that every variable X^i_t has as one parent some earlier version of it (X^i_{t-k}). I believe that assumptions (i) are probably of most interest since the assumptions required for (ii) are quite stringent and would be hard to tell in practice.

Although the extension of the identification proof of Peters to time-series may be fairly simple, I do not think this should count against the paper. The central idea of identifiable functional model class (from the Peters paper) - is both simple and profound. This paper represents an new application of this deep idea.
(e.g. noone argues that the concept of auto-regression is superfluous since it is a simple extension of regression!).

The paper also considers the problem of identifying structure from time-shifted series. This is a valuable contribution since such temporal shifting often arises in practical contexts due to time delays.







Summary: An interesting worthwhile paper that certainly improves upon existing time-series methods based on Granger causality.
Author Feedback

Author rebuttal: We thank all reviewers for their valuable input and would like to point out only few points:

- Theorem 1 (i) (our main result) does not require acyclicity of the summary time graph. Acyclicity is only one of two sufficient conditions.
- If the summary time graph is not required to be acyclic, it may happen that all nodes of the full time graph are directly connected. Then Theorem 1 (ii) does not hold.
- We agree that model checks have been used before. We will stress this point and include the references in the final version.
- We are currently working on a more formal statement of the correctness of the algorithm in Mooij et al 2009, which is too long to fit into the eight pages. We add a corresponding reference.
- Pruning parents is achieved by excluding those parents time series that are not required to obtain independent residuals.